# Validity of Pulmonary Valve *Z*-Scores in Predicting Valve-Sparing Tetralogy Repairs—Systematic Review [note 1]

**DOI:** 10.3390/children6050067

**Published:** 2019-05-04

**Authors:** Raina Sinha, Vasu Gooty, Subin Jang, Ali Dodge-Khatami, Jorge Salazar

**Affiliations:** 1Division of Pediatric and Congenital Cardiac Surgery, UT-Houston McGovern Medical School, Children’s Memorial Hermann Hospital, 6431 Fannin Street, MSB 6.264, Houston, TX 77030, USA; Ali.DodgeKhatami@uth.tmc.edu (A.D.-K.); Jorge.D.Salazar@uth.tmc.edu (J.S.); 2Division of Pediatric Cardiology, University of Texas Southwestern, Dallas Children’s Medical Center, 1935 Medical District Drive, Dallas, TX 75235, USA; vasu.gooty@gmail.com; 3Division of Pediatric Cardiac Surgery, University of Minnesota, Masonic Children’s Hospital, 2450 Riverside Ave, Minneapolis, MN 55454, USA; jangx147@umn.edu

**Keywords:** tetralogy of Fallot, valve sparing surgery, pulmonary valve *Z*-score, pulmonary stenosis, right ventricular outflow tract obstruction

## Abstract

There is a lack of consensus regarding the preoperative pulmonary valve (PV) *Z*-score “cut-off” in tetralogy of Fallot (ToF) patients to attempt a successful valve sparing surgery (VSS). Therefore, the aim of this study was to review the available evidence regarding the association between preoperative PV *Z*-score and rate of re-intervention for residual right ventricular outflow tract (RVOT) obstruction, i.e. successful valve sparing surgery. A systematic search of studies reporting outcomes of VSS for ToF was performed utilizing PubMed, EMBASE, and Scopus databases. Patients with ToF variants such as pulmonary atresia, major aortopulmonary collaterals, absent pulmonary valve, associated atrioventricular septal defect, and discontinuous pulmonary arteries were excluded. Out of 712 screened publications, 15 studies met inclusion criteria. A total of 1091 patients had surgery at a median age and weight of 6.9 months and 7.2 kg, respectively. VSS was performed on the basis of intraoperative PV assessment in 14 out of 15 studies. The median preoperative PV *Z*-score was −1.7 (0 to −4.9) with a median re-intervention rate of 4.7% (0–36.8%) during a median follow-up of 2.83 years (1.4–15.8 years). Quantitatively, there was no correlation between decreasing preoperative PV *Z*-scores and increasing RVOT re-intervention rates with a correlation coefficient of −0.03 and an associated *p*-value of 0.91. In observational studies, VSS for ToF repair was based on intraoperative evaluation and sizing of the PV following complete relief of all levels of obstruction of the RVOT, rather than pre-operative echocardiography derived PV *Z*-scores.

## 1. Introduction

Surgical repair of tetralogy of Fallot (ToF) often involves either a transannular patch (TAP) or valve sparing technique to relieve right ventricular outflow tract obstruction (RVOTO), with the potential need for a RV-PA conduit in cases with a coronary artery crossing the RVOT. In the past, the trigger to sacrifice the pulmonary valve (PV) and accept free pulmonary insufficiency (PI) had a lower threshold, whereas the modern era has shifted towards valve-sparing surgery (VSS) when possible. This strategy theoretically protects the right ventricle (RV) from chronic volume overload [1]. However, there are several controversies surrounding this approach, such as the ideal age and weight for complete repair, staged versus primary repair, the role of percutaneous interventions in staging, and the importance of sparing the RV infundibulum. PV annulus *Z*-score correlates, derived from preoperative transthoracic echocardiography measurements, have played a vital role in this decision-making process, and a score below −2 has been commonly referenced as a “cut-off” point for the insertion of a TAP [2]. Therefore, the aim of this study was to conduct a review of current literature to help identify practice patterns of VSS for ToF repair based on preoperative PV *Z*-scores, and analyze outcomes with respect to the incidence of subsequent re-intervention/reoperation for residual RVOT obstruction.

## 2. Materials and Methods

A comprehensive search of PubMed, Embase, and Scopus databases was conducted from their inception to April 2018, utilizing MeSH (Medical Subject Headings) terms “tetralogy of Fallot”. Only studies reporting results of ToF undergoing VSS were included. Those with ToF variants including pulmonary atresia, major aortopulmonary collateral arteries, absent pulmonary valve, atrioventricular septal defect, discontinuous pulmonary arteries (PA) or branch PA stenosis were excluded. Patient characteristics such as age, weight, follow-up duration, as well as pre-operative, intraoperative and post-operative details were reviewed for each publication. Our primary data point was the preoperative pulmonary valve *Z*-scores of patients undergoing VSS, and the secondary data point was re-intervention (surgical and catheterization) rates. We defined “re-intervention” as any procedure (surgical or catheter based), needed at the sub valvar, valvar, supra valvar, or distal branch PA levels. Those performed for residual shunts or complete heart block were omitted. Re-intervention rates were calculated by dividing the number of patients who underwent re-intervention by the total number of patients who underwent VSS, multiplied by a hundred. Values are reported in median with range or mean with standard deviation based on each publication’s available data. Microsoft Excel (version 2016, Microsoft, Redmond, WA, USA) was utilized to perform correlation and regression analysis.

## 3. Results

A thorough systematic literature search yielded a total of 712 publications which were identified and screened. After exclusion and cross-referencing, 15 studies that had VSS repair of ToF were deemed relevant and included in this systematic literature review (Figure 1).

The characteristics of these 15 eligible studies are summarized in Table 1 [1,3,4,5,6,7,8,9,10,11,12,13,14,15,16]. They include clinical outcomes from a diverse patient population from different continents spanning a time frame from 1997 to 2016. During this period, a total of 1091 patients underwent VSS for ToF. The median age at the time of surgical repair was 6.9 months (range 0.5 to 39.6 months), the median weight was 7.2 kg (range 3.4 to 11.7 kg), and the median follow-up was 2.83 years (range 1.4 to 15.8 years). The calculated median preoperative PV *Z*-score, in the studies that reported, it was −1.7 (range 0 to −4.9), with a median re-intervention rate of 4.7% (range 0% to 36.8%). Statistical analysis comparing the preoperative PV *Z*-score and RVOT re-intervention rates demonstrated no correlation with a correlation coefficient (r) of −0.03 and an associated *p*-value of 0.91.

A staged approach for VSS, via a modified Blalock–Taussig shunt (mBTs), RVOT stent, or patent ductus arteriosus (PDA) stent, occurred in seven studies, for a total of 63 patients (Table 2). Pulmonary valve morphology was described in nine out of the 15 studies, where the most common morphology observed was bicuspid, followed by tricuspid, and least commonly monocuspid (Table 2). Operative technique was reported in 14 out of the 15 studies, which included transatrial, transpulmonary, and transinfundibular approaches for the VSD repair as well the relief of RVOT obstruction (Table 1). The decision to proceed with VSS was made intraoperatively based on the assessment of the RVOT dimensions subsequent to relieving RVOT obstruction.

## 4. Discussion

Although a valve sparing surgical procedure for ToF repair can protect the RV from volume overload, the possibility of residual obstruction exists. The preoperative echocardiographic PV annulus *Z*-score, a value which is subject to variability and is operator dependent, has been utilized to guide the surgical decision between a TAP and VSS. However, there is inconsistency in its application as suggested by the variability of median *Z*-scores. Moreover, a “cut-off” PV *Z*-score value to delineate between the two surgical approaches has not been well established, with conflicting available literature. Some groups such as Awori et al. support a PV *Z*-score of −2 as a marker below which to insert a TAP [2]. In contrast, Stewart et al. observed that a PV *Z*-score greater than −4 was found to be a predictor of pulmonary valve preservation with minimal recurrent obstruction (*n* = 85) [3]. Ito et al. found that even in patients (*n* = 11) with PV *Z*-score < −4 (range −6.3 to −4.3), a valve sparing strategy was successful in 58% of cases [11].

Our review of 15 articles has revealed a median PV *Z*-score for the cohort of −1.7, with a wide range from 0 to −4.9. This refutes a “one size fits all” cut-off value, or a universal application of this echocardiographic measurement in ToF repairs. Approximately half of the articles had PV *Z*-score less than −2 (Figure 2), which suggests there are additional factors, when considering a potential VSS. The median re-intervention rate observed in our review of the 15 included studies was 4.7% (Figure 3). While there are few outliers, qualitatively there was no association between decreasing *Z*-scores and an increased rate of re-intervention as reflected by the correlation coefficient of −0.03. Notably, this data is reflective of short to mid–term results with a median follow-up of 2.8 years (range 1.4 to 15.8 years) along with associated inherent limitations/bias from single center retrospective studies, but it also indicates that the preoperative *Z*-score does not reliably predict valve sparing potential.

The surgical aim is to preserve adequate pulmonary valve “function” rather than simply the native leaflets, in order to minimize regurgitation and outflow obstruction. Valve morphology and effective orifice area are key determinants. Therefore, practice patterns based on the intraoperative assessment of the valve, rather than the echocardiography derived preoperative PV *Z*-score, seem to be the most cogent argument favoring VSS strategy [14]. The primary focus should be to address all levels of stenosis, beginning from the supravalvar area through a pulmonary arteriotomy, all the way down to the annulus. This was reported in 14 of the 15 studies describing their operative techniques. The valve is subsequently inspected and interventions performed to allow improved function of the native leaflets, involving techniques such as leaflet thinning and commissurotomies, to increase mobility of the free leaflet edges [11]. Extensive subvalvar resection and selective infundibular incisions are performed, if there continues to be subvalvar obstruction. At this point, the dimensions of the RVOT may be reassessed and if deemed appropriate, valve sparing repair performed. This has been our surgical approach with low rates of re-intervention, along with other centers as reported by Hickey et al. [14]. In their study, a total of 296 patients underwent VSS (preoperative median *Z*-score of −4.5), and over a median follow up period of 13.7 years, there was a re-intervention rate of 4.7% for recurrent RVOT stenosis.

Limitations of our analysis include the retrospective nature, surgical approaches spanning several decades, lack of uniform data points, limited follow up, as well as publication and selection biases of the reviewed articles. There was variability in reporting PV annulus sizes and RVOT gradients during the pre-operative, operative, and post-operative periods, along with disparities in the measurements of PV *Z*-scores, without a unified *Z*-score utilization across the articles. Therefore, the true effect of valve sparing surgery and defining it as a successful strategy could not be statistically evaluated in the short-, mid-, nor long-term outcomes for the entire cohort of 1091 patients. In addition, significant biases exist in patient selection, as well as surgeon bias for VSS.

In summary, the surgical approach towards ToF repair has shifted from accepting free pulmonary insufficiency by the liberal application of a TAP, towards performing a valve sparing operation whenever possible. This is evidenced by increasing reports of successful VSS for ToF with acceptable outcomes and low re-intervention rates for residual RVOT obstruction. Our literature review of >1000 patients revealed an overall low rate of re-intervention after VSS, with varying preoperative PV *Z*-scores. This implies that the preoperative PV annulus *Z*-score determined by echocardiography is not necessarily predictive of valve sparing potential and should not be the sole guide when deciding the fate of the native PV leaflets and annulus. Rather, intraoperative assessment of the valve size after complete surgical relief of all levels of obstruction is a better indicator, although no randomized control study to support this approach has been done. Furthermore, defining “valve sparing success” requires longer follow up and remains a grey philosophical zone rather than a clear cut definition.

## Figures and Tables

**Figure 1 children-06-00067-f001:**
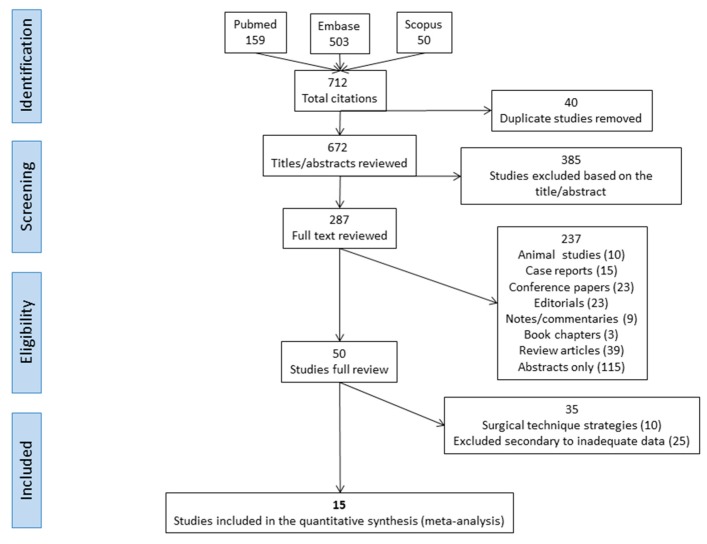
Flow diagram representing literature search and study selection process.

**Figure 2 children-06-00067-f002:**
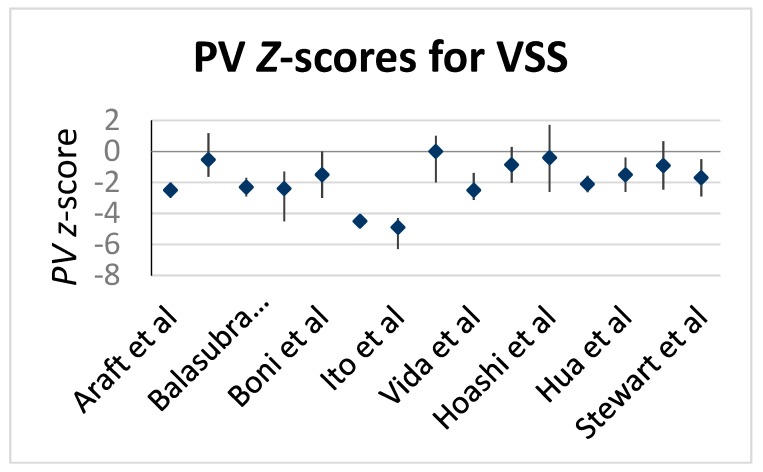
Shows the variation in the use of pre-operative PV *Z*-scores (mean/median) in the articles.

**Figure 3 children-06-00067-f003:**
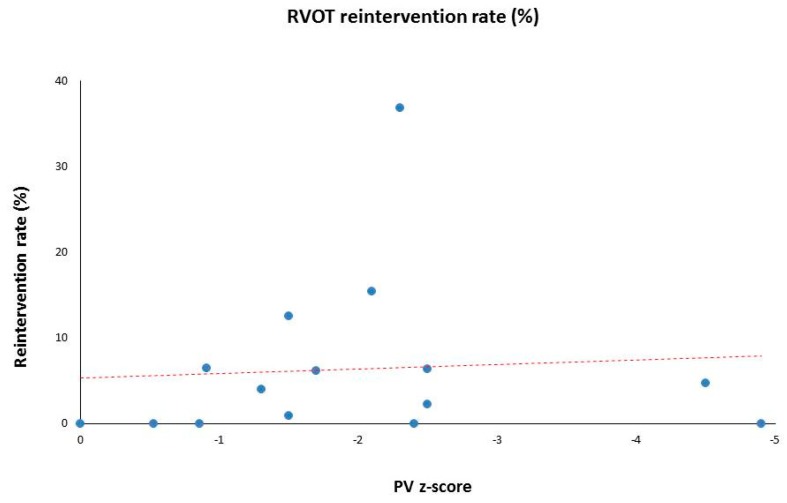
Low rate of re-intervention despite decreasing PV annulus *Z*-scores.

**Table 1 children-06-00067-t001:** Patient characteristics.

Author, Publication Year	Patients (*n*) with VSS	Median Age (months) at the Time of Surgery	Median (with Range) Weight at the Time of Surgery (kg)	Pre-Op PV *Z*-Score Median	RVOT Re-Intervention Rate (%)	Follow-Up (y) Median	Operative Technique
Stewart, 2005 [3]	82	9.4 ± 9 (mean)	7.4 ± 5.8 (mean)	−1.7	6.1	2.8	TA ± TP
Boni, 2009 [4]	24	8.1	8.05 (5-16.5)	−1.5	12.5	2.7	TA + TP
Hua, 2011 [5]	132	8.1 ± 3.2 (mean)	7.8 ± 6.2 (mean)	−1.5	0.9	2.3	TA + TP
Bove, 2012 [6]	48	7	-	−0.86	-	7.5	TA + TP
Vida, 2012 [7]	16	3.1	5.2 (4.6–6.8)	−2.5	6.3	1.4	TA + TP
Awori, 2013 [8]	46	6.5	6.6	−0.53	0	-	TP
Bautista-Hernandez, 2013 [9]	10	5.5	7.5 (4.7–47)	−2.4	0	1.8	TI + TP
Sasson, 2013 [10]	69	36	11.7 (4.3–49)	0	0	-	TA
Ito, 2013 [11]	11	6.9	(4.6–9.2)	−4.9	0	2.6	TA + TP
Hoashi, 2014 [12]	84	22.8 ± 16.8	9.3 ± 2.7	−1.3	4	15.8	TA + TP
Simon, 2017 [13]	46	4.8	6 (2–10)	−0.91	6.5	7.9	TA ± TI
Hickey, 2018 [14]	296	5.9	6.8 (2.5–85)	−4.5	4.7	13.7	TP
Hofferberth, 2018 [15]	162	3.2	5.4 (4.6–6.1)	−2.1	15.4	3	TI + TP
Arafat, 2018 [1]	46	11	9 (6–16)	−2.5	2.2	3.9	TA + TP
Balasubramanya, 2018 [16]	19	0.5	3.4 (2.5–3.9)	−2.3	36.8	2.2	-
**Median Values**	**Total *N* = 1091**	6.9	7.2	−1.7 (0 to −4.9)	4.5 (0–36.8)	2.8 (1.4–15.8)	

VSS—valve sparing surgery, PV—pulmonary valve, RVOT—right ventricular outflow tract, TA—transatrial, TP—transpulmonary, TI—transinfundibular.

**Table 2 children-06-00067-t002:** Pulmonary valve morphology and number of shunts.

Author, Year of Publication	Pulmonary Valve Morphology (%)			Prior Shunt (mBT Shunt or RVOT Stent or PDA Stent)
	Bicuspid	Tricuspid	Monocuspid	
Stewart, 2005 [3]	56/82 (68%)	26/82 (32%)	-	15 (18.3%)
Boni, 2009 [4]	15/24 (62.5%)	9/24 (37.5%)	-	0
Hua, 2011 [5]	99/111 (89.2%)	12/111 (10.8%)	-	0
Bove, 2012 [6]	-	-	-	5 (10%)
Bautista-Hernandez, 2013 [9]	8/10 (80%)	2/10 (20%)	-	0
Sasson, 2013 [10]	24.6%	-	-	2 (2.9%)
Ito, 2013 [11]	10/11 (90%)	-	-	0
Hoashi, 2014 [12]	56/84 (66.7%)	27/84 (32.1%)	-	11 (13.1%)
Hickey, 2018 [14]	-	-	-	13 (4.4%)
Hofferberth, 2018 [15]	123 (76%)	25 (15%)	14 (9%)	9 (5.6%)
Arafat, 2018 [1]	26 (56.5%)	17 (37%)	1 (2.2%)	8 (17.4%)
				**Total *N* = 63**

mBT—modified Blalock-Taussig shunt, PDA—patent ductus arteriosus.

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
