# Peer review of "Validity of Pulmonary Valve *Z*-Scores in Predicting Valve-Sparing Tetralogy Repairs—Systematic Review [Author-notes fn1-children-06-00067]"

_children, 2019, doi:10.3390/children6050067_

Round 1

Reviewer 1 Report

I would like to congratulate the authors in undertaking this extensive review. The findings are important for pediatric cardiology providers. The fact that the preoperative pulmonary valve Z score was a not a determinant of future re-interventions on these patients following valve sparing RVOT surgeries is not surprising, but an important piece of information. 

Author Response

Response to Reviewer 1 Comments

Point 1: I would like to congratulate the authors in undertaking this extensive review. The findings are important for pediatric cardiology providers. The fact that the preoperative pulmonary valve Z score was a not a determinant of future re-interventions on these patients following valve sparing RVOT surgeries is not surprising, but an important piece of information. 

Response 1: Thank you for the feedback and we concur that this is an important topic for discussion as our field has evolved tremendously in terms of having long term survivors.  Therefore when considering surgical options at the time of initial surgical repair, we must keep in mind their impact several decades later.  In the case of ToF repairs, the pendulum has swung from transannular patch repair to valve sparing whenever feasible and echocardiographic assessment of the pulmonary valve and RV outflow tract helps guide the surgical approach.  However, it is ultimately the intraoperative assessment of the valve which determines its fate.  Longer follow up of valve sparing ToF repairs is warranted to ultimately decide how much, if any residual stenosis is tolerated by the RV.  This would further guide the decision making process of having to choose between free pulmonary insufficieny or residual gradient across the RV outflow tract.

Reviewer 2 Report

The paper performed a systemic review of published studies to evaluate the validity of pulmonary valve Z scores in predicting valve-sparing tetralogy repairs. While the study is of potential importance, there are major limitations.

1.     The range of patient characteristics, such as age, weight, and follow-up time are very wide, while these can all potentially affect whether Z scores is a good predictor of success valve-sparing tetralogy repairs or not. In addition, the range of Z-score and re-intervention rate is wide. The wide range of these parameters reflects the big heterogeneity of these studies and raising the question of whether it is appropriate to perform systemic review with them. 

3.     VSS was performed on the basis of intraoperative PV assessment in 14 out of 15 studies, which can create selection bias.

4.     Result is qualitative but not quantitative. It will be more convincing if the author can provide correlation analysis even if the P value is not statistically significant.

Author Response

Response 1: Although it would be ideal to have homogenous data sets for review, these studies met our criteria in terms of reporting outcomes of valve sparing surgery (VSS) for ToF and therefore were included.  It is important to keep in mind that the decision of when and at what threshold to reintervene on the RVOT was not reported in all the studies, reflecting part of the heterogeneity amongst the articles.  Nonetheless, we concur with Reviewer 2 that one of the limitations is lack of uniformity amongst these studies.  As the body of literature continues to grow, perhaps a follow up review could be performed focusing on neonates or infants alone who underwent VSS for ToF in order to remove heterogeneity as a factor.

Response 2: As mentioned 14 of the 15 studies reported VSS outcomes based on pulmonary valve assessment intraoperatively.  This is reflective of the current methodology where the aim is to preserve the valve whenever feasible.  But the question remains, what are the parameters to guide this decision?  Should it be preoperative assessment based on Z scores or intraoperative assessment when a complete and detailed assessment of the RVOT at the valvar, supravalvar, and subvalvar levels can be performed? While it is true that selection bias is evident when performing intraoperative assessment of the PV, the echocardiographic measurements and thereby its Z score correlates, is also operator dependent and has its own inherent biases. Ultimately the aim of this discussion was to pose the question whether a preop measurement alone dictates the fate of the PV or not.

Response 3: Please note the edits in results section which demonstrated a correlation coefficient of -0.03 with a p-value of 0.91 when comparing the Z score and reintervention rates.  Therefore we can state that statistically we did not find a true correlation between the two values.

Reviewer 3 Report

The authors performed a systematic search of studies reporting outcomes of Valve-sparing Tetralogy Repairs (VSS) utilizing PubMed, EMBASE, and Scopus databases.  15 studies met inclusion criteria with a total of 1091 patients. VSS was performed on the basis of intraoperative PV assessment in 14 out of 15 studies. In most studies, VSS for ToF repair was based on intraoperative evaluation and sizing of the PV following complete relief of all levels of obstruction of the RVOT rather than pre-operative echocardiography derived PV Z-scores. The authors state that they found no correlation between decreasing preoperative echocardiographic PV Z scores and increasing RVOT re-intervention rates.

The criteria for valve sparing surgery in the repair of Tetralogy of Fallot is an important issue. The authors summarize 15 studies that deal with this topic.

The main message the reader gets is that the predictive value of the preoperative echocardiographic PV Z score is limited. No true correlation calculation is done, and it is unclear whether the patients who had the lowest Z score were more likely to undergo re-intervention. However, the studies clearly show that most patients with Z score of less than -2 who and VSS did not need re-intervention.

The data in table 1 is discussed in the discussion section rather than the results section.

As the authors note in the discussion, it is unclear what the Z score calculations were based on, thus it is hard to compare the different studies.

The RVOT pressure gradient post operatively, which is an important determinant of the decision to leave the valve, is not stated. This may correlate with the likelihood of re-intervention.

The authors found no correlation between Z score and re-intervention, however, they do not show the statistics.

There is no data on the preoperative echocardiographic PV Z scores of patients who ended up with trans anular patch.

Table 1, column 6, the title is Median RVOT reintervention (%). Only the last row is median.

The authors may try to explain the difference between studies with very low reintervention rates (0-6%) and very high reintervention rates (12.5% to 36.8%).

The authors may add the criteria for VSS intraoperatively, such as stated by Hickiy 2018 (Ref 14) “comparable to the risk of reoperation after TAP. If a 10-mm probe (z-score equivalent 1.2) can be passed after valvuloplasty, and the RV systolic pressure by needle manometry is less than 50 mm Hg, then the likelihood that an AS strategy will lead to risk of reoperation is low.”

Author Response

Response 1: Thank you for these comments.  We have added correlation and regression analysis to demonstrate that there is no correlation between Z score and reintervention rates with a correlation coefficient of -0.03 and a p-value of 0.91.

Data for patients with transannular patch rather than VSS was not reported amongst all the studies, and as we were focused on the VSS group, we did not include this data point in our results.

The term “median” has been removed from title of column 6 in Table 1.

In terms of reintervention, there is lack of explanation of the methodology in several papers and rather the data is simply stated such as the article by Balasubramanya et al.  There is no specific cut off value listed with regard to any echo gradients.  Boni et al noted 3 of the 24 patients had reintervention but the actual cause is only listed for two: residual RVOT muscle bundle, and branch PA stenosis.

There is wide variability in postoperative assessment of a successful VSS.  Some advocate an intraoperative needle measurement of RV/LV pressures when coming off CPB after the repair and proceed with VSS if the ratio is < 0.75 (Boni et al).  Others utilize RVOT gradient on TEE instead.  There does not appear to be a uniform cut off point amongst the studies reviewed.

Round 2

Reviewer 2 Report

The study has a lot of limitations but the they are adequately identified and acknowledged by the authors.